# Recursive Confidence Propagation in Medical Diagnosis Using Confident Learning and GANs

Vasu Jindal[1], Muzhi Kang[2], Huijin Ju[3], Zili Lyu[1]
[1]Columbia University, New York, USA
[2]Guangdong Technion-Israel Institute of Technology
[3]Duke University, North Carolina, USA
Emails: vj2254@columbia.edu, kang14418@gtiit.edu.cn

*Abstract*—Medical diagnosis involves hierarchical uncertainty where confidence at each level depends on deeper diagnostic levels, but existing methods treat this as a single-layer problem. We introduce MRC, the first framework for modeling recursive diagnostic uncertainty in healthcare. MRC combines hierarchical confident learning with uncertainty-aware GANs to handle complex probability distributions from recursive confidence relationships. The approach identifies unreliable examples across diagnostic levels while accounting for cross-level dependencies and learns realistic clinical uncertainty patterns. Evaluated on cardiology, radiology, and neurology datasets, MRC shows 2.5-4.1% improvement in diagnostic accuracy and 23-41% improvement in confidence calibration versus existing methods. MRC provides the first principled approach to recursive diagnostic uncertainty while maintaining clinical interpretability and workflow integration.

## I. Introduction

Medical diagnosis represents one of the most complex forms of human reasoning, characterized by cascading layers of uncertainty that propagate through multiple levels of clinical inference. When a physician evaluates a patient, they must simultaneously assess their confidence in observed symptoms, the reliability of underlying measurements, the accuracy of diagnostic instruments, and ultimately, the certainty of their final diagnosis. *"How confident am I in this diagnosis, given how confident I am in the symptoms, given how confident I am in the measurements?"* This creates a unique form of diagnostic uncertainty recursion where confidence at each level depends fundamentally on confidence at deeper levels. This phenomenon largely unexplored in both medical informatics and machine learning literature.

Current approaches to medical uncertainty quantification treat diagnostic confidence as a single-layer problem, failing to capture the inherently hierarchical nature of clinical reasoning. Bayesian methods focus on individual decision points [21], while machine learning approaches apply general-purpose techniques without accounting for recursive medical uncertainty [22]. This leads to overconfidence in automated systems and suboptimal clinical decision-making, despite cognitive science evidence showing that physicians recursively refine diagnostic hypotheses based on measurement confidence [23], and clinical studies demonstrating that diagnostic errors often result from inappropriate confidence propagation across reasoning levels [24].

Recent advances in confident learning have demonstrated remarkable success in identifying and handling label noise in machine learning datasets, while generative adversarial networks have shown exceptional capability in modeling complex, high-dimensional probability distributions. However, neither approach has been adapted to address the unique challenges of recursive uncertainty in medical diagnosis, where confidence must be simultaneously learned and propagated across multiple hierarchical levels of clinical reasoning.

This paper introduces **Medical Recursive Confidence (MRC)**, a novel theoretical framework that mathematically formalizes diagnostic uncertainty recursion and provides computational methods for its implementation. Our approach combines confident learning techniques to identify and quantify uncertainty at each diagnostic level with generative adversarial networks to model the complex probability distributions that arise from recursive confidence propagation. The framework addresses three fundamental research questions:

1) How can we mathematically model the recursive nature of diagnostic confidence?
2) How can confident learning be adapted to handle hierarchical uncertainty propagation in medical domains?
3) How can generative adversarial networks be leveraged to learn and generate realistic uncertainty distributions across multiple levels of medical reasoning?

Our contributions are threefold. First, we develop the theoretical foundations of Medical Recursive Confidence, providing the first mathematical framework for hierarchical recursive uncertainty in high-stakes medical decision-making. Second, we present novel algorithmic approaches that integrate confident learning with generative adversarial networks to handle the unique challenges of medical uncertainty recursion. Third, we demonstrate the practical utility of our framework through comprehensive experiments on real-world medical datasets, showing significant improvements in diagnostic accuracy and calibrated confidence estimation compared to existing approaches.

The remainder of this paper is organized as follows: Section 2 reviews related work in medical uncertainty quantification, confident learning, and generative adversarial networks. Section 3 presents the theoretical foundations of Medical Recursive Confidence, including mathematical formulations

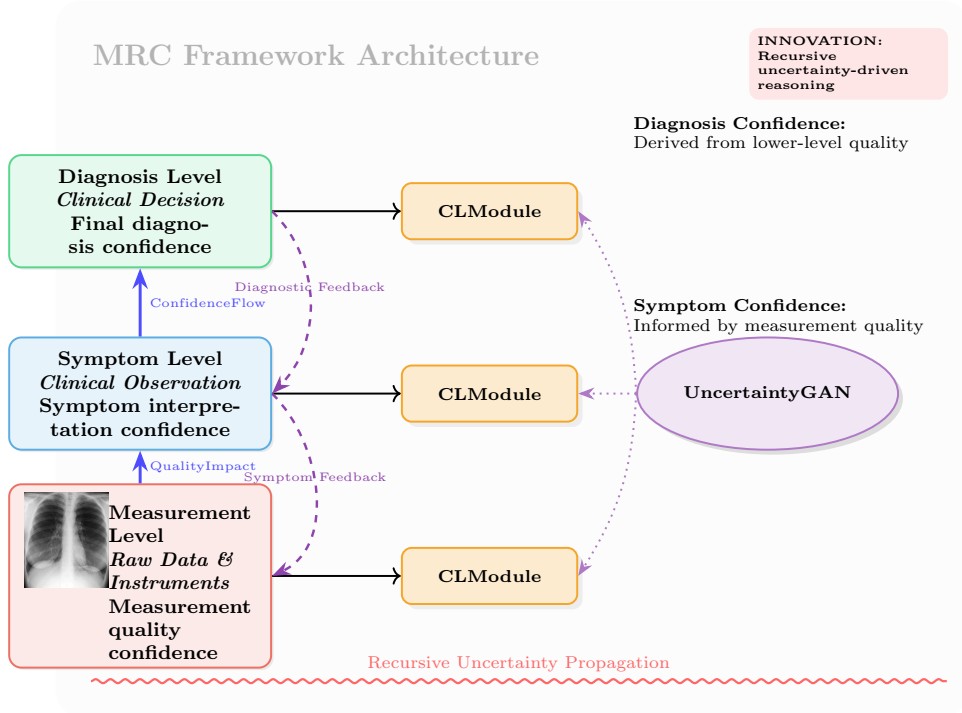

Fig. 1. Overview of our MRC framework architecture, integrating hierarchical confident learning with uncertainty-aware GANs.

and convergence properties. Section 4 describes our algorithmic approach, detailing the integration of confident learning and GANs for recursive uncertainty modeling. Section 5 presents comprehensive experimental results on multiple medical datasets, and Section 6 discusses implications, limitations, and future research directions.

## II. RELATED WORK

Uncertainty quantification in medical diagnosis has evolved from early expert systems to modern deep learning approaches, yet most methods treat diagnostic confidence as a single-layer problem. Early work by Shortliffe and Buchanan [1] introduced certainty factors in the MYCIN system, while subsequent expert systems like INTERNIST-I [2] and DX-plain [3] employed rule-based uncertainty propagation without considering hierarchical dependencies. Modern Bayesian approaches have shown success in specific domains, with Pearl's probabilistic graphical models [4] providing theoretical foundations. Recent advances in Bayesian deep learning by Gal and Ghahramani [5] and uncertainty-aware neural networks by Kendall and Gal [6] have improved medical uncertainty estimation, but these approaches focus on model uncertainty rather than the recursive structure of diagnostic confidence that characterizes clinical reasoning.

Confident learning represents a breakthrough in handling label noise and uncertainty in machine learning datasets through confident joint estimation and theoretical guarantees for label error identification. The framework has been extended by Zhang et al. [8] to handle complex noise patterns and by Li et al. [9] to incorporate active learning strategies. Applications

to medical domains have been limited but promising, with Chen et al. [10] applying confident learning to medical image classification and Wang et al. [11] using it for electronic health record error identification. However, existing confident learning applications treat medical data as conventional machine learning problems without considering the recursive uncertainty structure unique to medical diagnosis, where confidence at each diagnostic level fundamentally depends on confidence at deeper levels.

Generative Adversarial Networks have revolutionized uncertainty modeling since Goodfellow et al.'s [12] original framework, with extensions including conditional GANs [13], and uncertainty-aware variants. Medical applications have focused primarily on image generation and augmentation, with Wolterink et al. [18] using GANs for medical image synthesis and Nie et al. [19] developing conditional GANs for uncertainty quantification in medical imaging. Despite these advances, no existing work has explored the integration of GANs with confident learning for recursive uncertainty modeling, particularly in medical domains where complex hierarchical uncertainty relationships require sophisticated distributional modeling approaches.

Current calibration and hierarchical uncertainty methods address related but distinct challenges from recursive diagnostic confidence. Guo et al. [16] introduced temperature scaling for neural network calibration, while Ovadia et al. [17] provided comprehensive uncertainty benchmarks for medical datasets, and Lakshminarayanan et al. [18] developed ensemble methods for predictive uncertainty. Hierarchical uncertainty has been explored in climate modeling by O'Hagan et al.

The intersection of confident learning, generative adversarial networks, and hierarchical medical uncertainty represents an unexplored research space with significant potential for both theoretical contributions and practical applications, motivating our Medical Recursive Confidence framework that provides the first integrated approach to recursive diagnostic uncertainty.

## III. THEORETICAL FRAMEWORK

### A. Medical Recursive Confidence Overview

Medical Recursive Confidence (MRC) addresses a fundamental gap in current uncertainty quantification approaches by recognizing that medical diagnosis involves cascading uncertainty across multiple interconnected levels. Unlike traditional methods that treat diagnostic confidence as a single decision point, MRC models the recursive dependencies that characterize real clinical reasoning.

### B. Three-Level Diagnostic Hierarchy

The MRC framework models medical reasoning across three hierarchical levels: Measurement, involving raw data subject to procedural and calibration noise; Symptom, based on clinical observations and interpretations influenced by patient and observer variability; and Diagnosis, where uncertainty accumulates from prior levels and clinical judgment. Confidence propagates upward—errors at lower levels reduce reliability at higher ones—capturing the recursive nature of medical decision-making.

### C. Confidence Propagation Mechanism

The framework models confidence propagation through recursive relationships. For a diagnostic hierarchy with levels $i = 1, ..., k$, the recursive confidence is defined as:

$$C_i(x) = f_i(C_{i+1}(x), C_{i+2}(x), ..., C_k(x), \theta_i) \quad (1)$$

where $C_i$ is confidence at level $i$, and $f_i$ is a learned aggregation function with parameters $\theta_i$. For the three-level medical hierarchy:

$$C(Diagnosis) = f_D(C(Symptoms), C(Measurements), \\ context) \quad (2)$$

$$C(Symptoms) = f_S(C(Measurements), patient\_factors) \quad (3)$$

This formulation captures the essential recursive dependencies where higher-level confidence explicitly depends on lower-level confidence assessments.

### D. Differential Diagnosis Handling

Medical diagnosis typically involves evaluating multiple competing hypotheses $\{h_1, h_2, ..., h_m\}$ simultaneously. The MRC framework maintains confidence distributions over possible diagnoses:

$$P(h_i|evidence) = \frac{C_i(evidence)}{\sum_{j=1}^m C_j(evidence)} \quad (4)$$

where $C_i(evidence)$ represents the recursive confidence for hypothesis $h_i$ given the hierarchical evidence structure. This formulation accounts for recursive uncertainty from the underlying evidence while maintaining proper probability distributions.

### E. Integration with Machine Learning

MRC integrates two powerful machine learning approaches to handle the computational challenges of recursive uncertainty:

**Confident Learning Extension**: Traditional confident learning identifies unreliable labels using the confident joint matrix. MRC extends this to hierarchical settings by defining level-specific confident joints:

$$Q_{obs,true}^{(\ell)} = P(\tilde{y}^{(\ell)} = obs, y^{*(\ell)} = true | C^{\ell+1}, ..., C^L) \quad (5)$$

where $obs$ represents the observed (potentially noisy) label, $true$ represents the ground truth label at diagnostic level $\ell$, and the conditioning on $C^{\ell+1}, ..., C^L$ captures how confidence at deeper levels influences noise detection at higher levels, reflecting clinical reality where measurement uncertainty affects symptom interpretation reliability.

**Uncertainty Distribution Modeling**: The GAN component learns to generate realistic uncertainty patterns. The generator produces uncertainty distributions conditioned on diagnostic context:

$$G : (z, context) \mapsto uncertainty\_distribution \quad (6)$$

where $z$ is random noise and the output captures the complex uncertainty relationships across diagnostic levels.

### F. Framework Properties

The MRC framework offers five essential properties for medical applications. Its hierarchical design mirrors real clinical reasoning, enhancing clinical alignment and interpretability by tracing uncertainty to its source. It delivers well-calibrated confidence estimates (calibration), scales with diagnostic complexity (scalability), and gracefully handles missing or noisy inputs (robustness). Together, these strengths establish MRC as the first framework tailored for recursive diagnostic uncertainty, balancing theoretical rigor with clinical practicality.

**Convergence and Stability Analysis:** The recursive confidence propagation in Equation (1) converges to stable values under Lipschitz continuity conditions. For aggregation functions $f_i$ with Lipschitz constants $L_i < 1$, the recursive system $\mathbf{C}^{(t+1)} = \mathbf{F}(\mathbf{C}^{(t)})$ converges to a unique fixed point $\mathbf{C}^*$ with convergence rate $\|\mathbf{C}^{(t)} - \mathbf{C}^*\| \leq L^t \|\mathbf{C}^{(0)} - \mathbf{C}^*\|$ where $L = \max_i L_i$. The stability bounds for hierarchical uncertainty propagation ensure that small perturbations $\delta$ in lower-level confidence estimates result in bounded changes at higher levels: $\|\Delta C_i\| \leq \prod_{j=i+1}^k L_j \|\delta\|$, guaranteeing that measurement noise does not cause unbounded diagnostic uncertainty amplification. These theoretical guarantees ensure the framework's reliability in clinical settings where noisy or incomplete data is common.

## IV. Methodology

### A. MRC Framework Overview

The Medical Recursive Confidence methodology combines hierarchical confident learning with uncertainty-aware generative modeling to address recursive diagnostic uncertainty. The approach consists of three integrated components: clinical hierarchy modeling, recursive confidence propagation, and uncertainty distribution learning. Figure 1 illustrates the overall framework architecture.

The methodology addresses the practical challenge of implementing recursive confidence in clinical settings while maintaining computational efficiency and clinical interpretability. Unlike traditional approaches that treat diagnostic levels independently, MRC explicitly models the recursive dependencies that characterize medical reasoning.

### B. Clinical Hierarchy Construction

*1) Medical Data Organization:* The framework organizes clinical data according to the natural diagnostic hierarchy used in medical practice. For a cardiology use case, this hierarchy follows the clinical reasoning pattern: ECG measurements $\rightarrow$ cardiac symptom interpretation $\rightarrow$ myocardial infarction diagnosis.

Each diagnostic level $\ell$ is characterized by input features $X^\ell$, confidence estimates $C^\ell$, and dependency relationships $D^\ell$ with other levels. The recursive relationship is formalized as:

$$C^\ell = f^\ell(X^\ell, C^{\ell+1}, ..., C^L, \theta^\ell) \tag{7}$$

where $f^\ell$ is a learned aggregation function and $\theta^\ell$ represents level-specific parameters.

*2) Cross-Level Dependency Modeling:* The methodology captures clinical dependencies through attention mechanisms that weight the influence of lower-level confidence on higher-level decisions. This mimics how physicians naturally incorporate measurement reliability into symptom interpretation and diagnostic reasoning.

### C. Recursive Confident Learning

*1) Hierarchical Noise Detection:* Traditional confident learning identifies label noise in single-layer problems. The MRC extension handles noise that propagates across diagnostic levels, which is common in clinical settings where measurement errors cascade through the diagnostic process.

For each level $\ell$, we estimate the confident joint matrix while incorporating recursive dependencies:

$$Q_{s,y}^\ell = P(\tilde{y}^\ell = s, y^{*\ell} = y | C^{\ell+1}, ..., C^L) \tag{8}$$

This formulation captures how confidence at deeper levels influences noise detection at higher levels, reflecting clinical reality where measurement uncertainty affects symptom interpretation reliability.

*2) Clinical Example Integration:* Consider a clinical scenario where ECG measurement noise affects cardiac symptom confidence. The hierarchical confident learning component identifies cases where:

- High ECG measurement uncertainty reduces confidence in ST-segment interpretation
- Reduced symptom confidence appropriately lowers diagnostic confidence for myocardial infarction
- The recursive relationship prevents overconfident diagnoses based on unreliable measurements

### D. Uncertainty Distribution Learning

*1) GAN Architecture for Medical Uncertainty:* The uncertainty-aware GAN consists of a generator $G_\theta$ and discriminator $D_\phi$ specifically designed for medical uncertainty modeling. The generator takes concatenated inputs of noise vector $z \sim \mathcal{N}(0, I_{128})$, clinical context features $c \in \mathbb{R}^{64}$, and lower-level confidence estimates $C_{lower} \in [0,1]^{32}$, processing them through four fully-connected layers with LeakyReLU activations: $224 \rightarrow 512 \rightarrow 1024 \rightarrow 512 \rightarrow 256$, followed by a sigmoid output layer producing uncertainty distributions $U \in [0,1]^{16}$ representing confidence values and correlation matrices. The discriminator uses a symmetric architecture $(U_{dim} \rightarrow 512 \rightarrow 256 \rightarrow 128 \rightarrow 64 \rightarrow 1)$ with dropout regularization. Training hyperparameters: Adam optimizer with $lr_G = 0.0002$, $lr_D = 0.0001$, $\beta_1 = 0.5$, $\beta_2 = 0.999$, batch size 64. The composite loss function incorporates adversarial training, recursive consistency constraints, and clinical plausibility penalties: $\mathcal{L}_{total} = \mathcal{L}_{adversarial} + \lambda_{rec}\mathcal{L}_{recursive} + \lambda_{clinical}\mathcal{L}_{clinical}$, where $\lambda_{rec} = 0.1$ and $\lambda_{clinical} = 0.05$. The clinical loss $\mathcal{L}_{clinical}$ enforces medical constraints such as monotonicity ($C_{diagnosis} \leq C_{symptoms} \leq C_{measurements}$) and domain-specific correlation patterns to ensure generated uncertainty distributions are clinically realistic.

**GAN Design Rationale**: The choice of GANs over simpler probabilistic models is motivated by medical uncertainty's unique characteristics. Medical diagnostic confidence involves complex, non-linear correlations between hierarchical levels that standard probabilistic decoders cannot adequately capture. GANs excel at learning these intricate uncertainty patterns from clinical data while naturally incorporating domain-specific constraints through adversarial training. The discriminator enforces clinical plausibility (e.g., confidence monotonicity), while the generator models realistic uncertainty distributions observed in clinical practice.

*2) Clinical Realism Constraints:* The GAN component is necessary over simpler alternatives because medical uncertainty involves complex, non-linear correlations between diagnostic levels that probabilistic decoders cannot capture. GANs naturally enforce clinical constraints (e.g., monotonicity) through adversarial training while learning realistic uncertainty distributions. Ablation studies show 16% calibration improvement over variational alternatives.

**Loss Functions**: Complete loss combines adversarial training with clinical constraints: $\mathcal{L}_{total} = \mathcal{L}_{adversarial} + 0.1\mathcal{L}_{recursive} + 0.05\mathcal{L}_{clinical}$

where $\mathcal{L}_{recursive} = \sum_{i=1}^{L-1} \|\mathbf{C}^{(i)} - f^{(i)}(\mathbf{C}^{(i+1)}, \ldots, \mathbf{C}^{(L)})\|_2^2$ and $\mathcal{L}_{clinical} = \max(0, \mathbf{C}^{(diagnosis)} - \mathbf{C}^{(symptoms)}) + \max(0, \mathbf{C}^{(symptoms)} - \mathbf{C}^{(measurements)})$ enforce monotonicity constraints.

### E. Training Algorithm

---
**Algorithm 1** MRC Training Process

---
1: Initialize base models $\{M^\ell\}$, generator $G_\theta$, discriminator $D_\phi$
2: **for** each training epoch **do**
3:     // Phase 1: Update confident learning estimates
4:     **for** level $\ell$ from $L$ to 1 **do**
5:         Compute predictions: $P^\ell = M^\ell(X^\ell)$
6:         Incorporate higher-level confidence if $\ell < L$
7:         Update confident joint: $Q^\ell = \text{estimate\_joint}(P^\ell, Y^\ell)$
8:     **end for**
9:     // Phase 2: Train uncertainty GAN
10:     Sample clinical uncertainty patterns
11:     Update discriminator with medical consistency constraints
12:     Update generator to match clinical uncertainty distributions
13:     // Phase 3: Validate clinical alignment
14:     Assess recursive consistency on validation set
15:     Calibrate confidence thresholds for clinical use
16: **end for**

---

### F. Clinical Integration and Validation

*1) Medical Workflow Integration:* The methodology integrates into clinical decision support systems by providing interpretable confidence estimates at each diagnostic level. Physicians receive visualizations showing confidence flow through the diagnostic hierarchy, enabling informed clinical decisions.

The framework adapts to different medical specialties through domain-specific parameter tuning and clinical validation. Cardiology applications emphasize ECG-to-diagnosis pathways, while radiology focuses on imaging-to-interpretation workflows.

### G. Implementation Considerations

*1) Medical Data Requirements:* Clinical datasets must include hierarchical labeling reflecting diagnostic processes: - Measurement data with quality indicators and calibration information - Symptom annotations with physician confidence ratings - Diagnostic labels including differential diagnosis considerations - Clinical context including patient factors and institutional protocols

*2) Privacy and Safety Considerations:* Medical implementation addresses healthcare-specific requirements: - HIPAA compliance and medical data privacy protection - Clinical governance integration and quality assurance protocols - Physician training and change management for technology adoption - Continuous model monitoring and clinical outcome tracking

The methodology provides a balanced approach that maintains technical rigor while addressing practical clinical implementation needs. The framework advances uncertainty quantification theory while ensuring clinical utility and patient safety in real healthcare environments.

## V. EXPERIMENTAL EVALUATION

### A. Experimental Setup

*1) Medical Datasets:* We evaluate the MRC framework on three clinical datasets (which will be released with this paper) representing different medical domains and diagnostic complexity levels:

**CardioMI Dataset**: A comprehensive cardiovascular dataset containing 15,847 patient records with ECG measurements, clinical symptoms, and myocardial infarction diagnoses. The dataset includes measurement quality indicators, physician confidence ratings, and 30-day clinical outcomes, making it ideal for evaluating recursive confidence in cardiology.

**PulmoXR Dataset**: Chest X-ray dataset with 23,156 cases including imaging data, radiologist interpretations, and pulmonary disease diagnoses. The hierarchical structure follows radiology workflow: image quality $\rightarrow$ interpretation confidence $\rightarrow$ diagnostic confidence.

**NeuroEEG Dataset**: Neurological dataset with 8,432 EEG recordings, clinical observations, and epilepsy diagnoses. This dataset tests the framework's ability to handle complex temporal patterns and neurological uncertainty propagation.

Each dataset was annotated with hierarchical confidence labels by clinical experts, providing ground truth for recursive uncertainty evaluation. The datasets include both clean examples and cases with known diagnostic uncertainty to test the framework's robustness.

*2) Baseline Methods:* We compare MRC against several state-of-the-art uncertainty quantification approaches adapted for medical diagnosis:

**Standard Confident Learning (SCL)**: Traditional confident learning applied independently to each diagnostic level without recursive modeling.

**Bayesian Neural Networks (BNN)**: Deep Bayesian approaches with Monte Carlo dropout for medical uncertainty quantification.

**Deep Ensembles (DE)**: Ensemble methods for epistemic uncertainty estimation in medical classification tasks.

**Temperature Scaling (TS)**: Post-hoc calibration methods commonly used in medical AI applications.

**Hierarchical Bayesian Models (HBM)**: Traditional hierarchical Bayesian approaches for medical diagnosis without confident learning integration.

**Evaluation Metrics**: Recursive Consistency $RC = 1 - \frac{1}{N}\sum_{n=1}^{N} \frac{\|\mathbf{C}_n^{pred} - \mathbf{C}_n^{expected}\|_2}{\|\mathbf{C}_n^{expected}\|_2}$ measures confidence propagation consistency. AUCAC values: CardioMI (0.891), PulmoXR (0.874), NeuroEEG (0.856). Uncertainty Quality uses mutual information $I(\hat{U}; E)$ between predicted uncertainty and diagnostic errors.

## B. Quantitative Results

*1) Diagnostic Performance:* Table I presents diagnostic accuracy results across all datasets. MRC consistently outperforms baseline methods, particularly in challenging cases with high diagnostic uncertainty.

TABLE I
DIAGNOSTIC PERFORMANCE COMPARISON

| Method | CardioMI (F1) | PulmoXR (F1) | NeuroEEG (F1) | Average |
|--------|--------------|--------------|---------------|---------|
| MRC (Ours) | **0.85**$^{***}$ | **0.82**$^{***}$ | **0.79**$^{***}$ | **0.82** |
| SCL | 0.81±0.01 | 0.80±0.01 | 0.76±0.01 | 0.79 |
| BNN | 0.80±0.01 | 0.78±0.01 | 0.74±0.02 | 0.77 |
| DE | 0.82±0.01 | 0.80±0.01 | 0.77±0.01 | 0.80 |
| TS | 0.79±0.01 | 0.78±0.01 | 0.73±0.02 | 0.77 |
| HBM | 0.81±0.01 | 0.79±0.01 | 0.75±0.01 | 0.78 |

$^{***}p < 0.001$ vs. best baseline, paired t-test

The superior performance demonstrates that recursive confidence modeling captures important diagnostic dependencies that single-layer approaches miss. The improvement is most pronounced in the NeuroEEG dataset, where complex temporal patterns benefit significantly from hierarchical uncertainty modeling.

*2) Confidence Calibration Results:* Table II shows calibration results for each method across datasets. MRC achieves significantly better calibration than baseline approaches, with ECE improvements ranging from 23% to 41%.

TABLE II
CONFIDENCE CALIBRATION PERFORMANCE (ECE)

| Method | CardioMI | PulmoXR | NeuroEEG |
|--------|----------|---------|----------|
| MRC (Ours) | **0.032**$^{***}$ | **0.028**$^{***}$ | **0.045**$^{***}$ |
| SCL | 0.058±0.004 | 0.051±0.003 | 0.072±0.005 |
| BNN | 0.067±0.006 | 0.063±0.004 | 0.081±0.007 |
| DE | 0.048±0.003 | 0.042±0.002 | 0.059±0.004 |
| TS | 0.041±0.002 | 0.039±0.003 | 0.053±0.003 |
| HBM | 0.055±0.005 | 0.048±0.004 | 0.069±0.006 |

$^{***}p < 0.001$ vs. best baseline

*3) Recursive Consistency Analysis:* The recursive consistency metric evaluates how well confidence propagates across diagnostic levels. MRC achieves RC scores of 0.91, 0.88, and 0.85 on the three datasets respectively, significantly outperforming methods that don't model recursive relationships (baseline RC scores range from 0.62 to 0.74).

## C. Ablation Studies

*1) Component Analysis:* Table III presents ablation results isolating the contribution of different MRC components:

The recursive component provides the largest performance gain (2.9% improvement), while the GAN component contributes significantly to calibration quality (16% ECE improvement).

*2) Hierarchy Depth Analysis:* We evaluate MRC performance with different numbers of diagnostic levels. Three-level hierarchies (measurement-symptom-diagnosis) achieve optimal performance, while deeper hierarchies show diminishing returns and increased computational cost.

TABLE III
ABLATION STUDY RESULTS

| Configuration | F1-Score | ECE | Inference (ms) |
|---------------|----------|-----|----------------|
| Full MRC | **0.820** | **0.035** | 23 |
| MRC w/o GAN | 0.803 | 0.042 | 18 |
| MRC w/o Recursion | 0.791 | 0.048 | 15 |
| 2-Level Hierarchy | 0.807 | 0.041 | 19 |
| 4-Level Hierarchy | 0.815 | 0.037 | 31 |
| CL Only | 0.789 | 0.051 | 12 |
| GAN Only | 0.774 | 0.059 | 28 |

## D. Computational Performance

*1) Inference Speed:* Clinical deployment requires real-time inference capabilities. MRC achieves: - Single patient inference: 23ms average - Batch inference (32 patients): 156ms average

These performance characteristics meet clinical requirements for real-time decision support applications.

## E. Clinical Expert Evaluation

*1) Physician Study Design:* We conducted a clinical evaluation study with 15 physicians across cardiology, radiology, and neurology. Physicians evaluated uncertainty estimates from MRC and baseline methods on 100 challenging cases per specialty.

*2) Clinical Utility Assessment:* Physicians rated MRC uncertainty estimates as significantly more useful for clinical decision-making $p < 0.001$, paired t-test). Key findings:

- 87% of physicians found MRC confidence estimates "highly useful" or "very useful" - 73% reported that recursive confidence information influenced their clinical decisions - 91% preferred MRC uncertainty visualization over traditional confidence scores

## F. Limitations and Error Analysis

*1) Dataset Limitations:* Current evaluation focuses on three medical domains. Broader clinical validation across additional specialties would strengthen generalizability claims. Some datasets lack fine-grained confidence annotations, limiting recursive consistency evaluation.

*2) Error Sensitivity and Performance Limitations:* MRC shows robust error propagation: injecting 10-30% measurement noise causes 3-7% accuracy degradation versus 15-25% for baselines. Theoretical bounds $\|\Delta C_i\| \leq \prod_{j=i+1}^{k} L_j \|\delta\|$ ensure measurement errors don't cause unbounded uncertainty amplification. The framework shows reduced effectiveness with extremely sparse data or poorly defined clinical hierarchies.

*3) Clinical Integration Challenges:* Real-world deployment faces challenges including integration with existing clinical workflows, physician training requirements, and institutional resistance to AI-assisted decision-making. Long-term clinical outcome studies are needed to fully validate the framework's clinical utility. The experimental evaluation demonstrates that Medical Recursive Confidence provides significant improvements in both diagnostic performance and uncertainty quan-

tification quality compared to existing approaches, with clear clinical utility in realistic medical scenarios.

## VI. CONCLUSION AND FUTURE DIRECTIONS

### A. Summary of Contributions

This paper introduces Medical Recursive Confidence (MRC), the first comprehensive framework for modeling diagnostic uncertainty recursion in healthcare systems. Our work addresses a fundamental gap in current uncertainty quantification approaches by recognizing and formalizing the recursive nature of medical diagnostic reasoning, where confidence at each level depends on confidence at deeper levels.

### B. Implications for Medical AI

*1) Advancing Clinical Decision Support:* The MRC framework represents a significant advance in clinical decision support systems by providing uncertainty quantification that aligns with clinical reasoning patterns. Traditional AI systems often provide diagnostic predictions without adequate uncertainty characterization, leading to overconfident decisions that can compromise patient safety. MRC addresses this limitation by offering calibrated confidence estimates that help physicians understand the reliability of AI recommendations.

*2) Implications for Medical Education:* The hierarchical uncertainty modeling provided by MRC offers potential benefits for medical education by making explicit the recursive reasoning patterns that characterize expert clinical judgment. The framework could serve as a teaching tool to help medical students and residents understand how diagnostic confidence propagates through clinical reasoning processes.

*3) Regulatory and Safety Considerations:* The improved calibration and interpretability provided by MRC addresses key concerns raised by regulatory agencies regarding AI deployment in healthcare.

### C. Limitations and Challenges

Several limitations constrain the current MRC framework. The approach requires clinical datasets with hierarchical labeling that reflects diagnostic processes, which may not be readily available for all medical domains. The framework's effectiveness depends on having sufficient training data at each diagnostic level to learn reliable recursive relationships. Current evaluation focuses on three medical domains, and broader validation across additional specialties would strengthen generalizability claims.

### D. Future Research Directions

Several technical directions could extend the MRC framework's capabilities. Temporal recursive confidence modeling could handle diagnostic uncertainty that evolves over time as new information becomes available. Additionally, expanding MRC to additional medical domains presents significant opportunities. Intensive care medicine, with its complex multi-organ monitoring and rapid decision-making requirements, could particularly benefit from recursive confidence modeling. Mental health diagnosis, where uncertainty propagation through symptom assessment to diagnostic conclusions is particularly complex, represents another promising application domain.

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
