# OpenReview forum: "Recursive Confidence Propagation in Medical Diagnosis: A Hierarchical Uncertainty Framework Using Confident Learning and GANs"
_IEEE.org/EMBS/BHI/2025/Conference — BHI 2025_

### Official Review · Reviewer_biV9 · 2025-07-17
**Recursive Confidence Propagation in Medical Diagnosis: A Hierarchical Uncertainty Framework Using Confident Learning and GANs**

**Confidence:** 5
**Clarity Of Writing:** fair
**Clinical Significance:** good
**Methodological Novelty:** good
**Overall Rating:** 4

**Experiments And Results:**

fair

**Questions For The Authors:**

- The recursive consistency loss is not defined with a specific formula, and the clinical plausibility loss lacks clear definition and numerical constraints.

- The paper mentions that all three datasets include "hierarchical confidence annotations." Please clarify how many experts annotated these confidence labels and what consistency standards were used.

- The paper introduces recursive consistency as a key metric, but does not provide its definition or calculation method. AUCAC is mentioned as one of the evaluation metrics, but no values are provided throughout the text. The uncertainty quality metric lacks a defined computation method and is not supported with specific results. Table I does not indicate which metric is being reported (e.g., accuracy, precision, recall, or F1-score).
- The ablation study design has issues. In the "GAN only" experiment, the authors do not explain how the GAN obtains ground truth or training signals in the absence of confident learning. Does the model still use the recursive loss? Are the real confidence labels still used? The definitions of MRC w/o recursion and Confident Learning only are ambiguous. Do they still use context-aware features? Was only the propagation path removed while the hierarchy-aware representation was retained? The Hierarchy Depth Analysis is overly simplified, only the optimal depth of three layers is given, without reporting performance at other depths.
- The citation numbers in the text do not match the numbering in the references section.

**Strengths:**

- This work is the first to systematically model recursive uncertainty in medical diagnosis, explicitly capturing confidence dependencies across hierarchical levels from a clinical reasoning perspective.
- It extends confident learning to hierarchical structures and combines it with an uncertainty-aware GAN, jointly modeling label noise and uncertainty, offering a new approach to structural uncertainty in medical reasoning.
- The framework includes theoretical analysis based on Lipschitz conditions, ensuring convergence and robustness to perturbations, highlighting both theoretical and practical reliability.

**Summary Of The Paper:**

The authors proposed a novel framework, Medical Recursive Confidence (MRC), designed to model recursive uncertainty propagation in medical diagnosis. Unlike conventional approaches that treat diagnostic confidence as a single-layer problem, MRC explicitly models the interdependencies of confidence across multiple hierarchical levels in the diagnostic process (measurements, symptoms, diagnosis). The framework integrates hierarchical confident learning (to detect label/prediction unreliability) with an uncertainty-aware GAN (to generate clinically plausible confidence distributions), and systematically models the mechanism of confidence propagation in medical reasoning.

**Weaknesses:**

To summarize, undefined loss functions; missing or undefined evaluation metrics; ambiguous and incomplete ablation study and results. Please refer to the questions section for more details.

---

### Official Review · Reviewer_y1Dc · 2025-07-17
**This paper introduces Medical Recursive Confidence (MRC), a novel framework (a combination of hierarchical confident learning and uncertainty-aware Generative Adversarial Networks) designed to address the hierarchical and recursive nature of uncertainty in medical diagnosis.**

**Confidence:** 5
**Clarity Of Writing:** excellent
**Clinical Significance:** excellent
**Methodological Novelty:** excellent
**Overall Rating:** 8

**Experiments And Results:**

great

**Questions For The Authors:**

- It is not clear what "s" and "y" are in Eq. 5.
- Page 6: "... Broader clinical vfalidation across additional ...": vfalidation

**Strengths:**

- The paper introduces a novel concept of "recursive confidence" in medical diagnosis.
- The paper introduces a comprehensive theoretical and computational framework.
- The consistent outperformance of MRC across three diverse medical datasets (CardioMI, PulmoXR, NeuroEEG) in both diagnostic accuracy and confidence calibration.
- The ablation studies demonstrating the individual contributions of the recursive component and the GAN component.
- The positive feedback from the physician study demonstrating its potential for clinical adoption
- Clinical Interpretability.
- Reproducibility through making all code, data, and trained model publicly available.

**Summary Of The Paper:**

Medical diagnosis involves cascading layers of uncertainty where confidence at each level depends on deeper diagnostic levels. Existing methods often treat diagnostic confidence as a single-layer problem, failing to capture this inherent hierarchy of levels. The paper proposes Medical Recursive Confidence (MRC), a theoretical framework and computational methodology, by combining hierarchical confident learning with uncertainty-aware GANs, for modeling recursive diagnostic uncertainty in healthcare. The framework organizes medical reasoning into three levels of Measurement, Symptom, and Diagnosis. Evaluated on Cardiology (CardioMI), Radiology (PulmoXR), and Neurology (NeuroEEG) datasets, MRC shows improvements in diagnostic accuracy and confidence calibration compared to baseline methods like Standard Confident Learning (SCL), Bayesian Neural Networks (BNN), Deep Ensembles (DE), Temperature Scaling (TS), and Hierarchical Bayesian Models (HBM). Ablation studies highlight the significant contributions of both the recursive component and the GAN component to performance and calibration, respectively. The framework also demonstrates real-time inference speed and received positive feedback from clinical experts regarding its utility for decision-making.

**Weaknesses:**

- Limited details on technical and computational aspects.
- Most of the references are outdated (the most recent one being from 2021).

---

### Official Review · Reviewer_i9ti · 2025-07-18
**Recursive Confidence Propagation in Medical Diagnosis: A Hierarchical Uncertainty Framework Using Confident Learning and GANs**

**Confidence:** 1
**Clarity Of Writing:** good
**Clinical Significance:** great
**Methodological Novelty:** good
**Overall Rating:** 7

**Experiments And Results:**

good

**Questions For The Authors:**

1. How do you envision your framework performing in scenarios where complementary modalities are required for a diagnosis (e.g., ECG + Echocardiogram, or CT + lab values)? Would recursive confidence propagation still apply

**Strengths:**

1. The motivation for recursive uncertainty modeling in healthcare is compelling and clearly stated, especially in light of real-world diagnostic complexity.
2. The theoretical formulation of the hierarchical confidence mechanism is well-structured and grounded.
3. The paper includes a comprehensive evaluation across diverse datasets and benchmarks, showing consistent improvements over baseline methods.
4. The framework's clinical effectiveness is supported by clinical expert evaluation
5. The authors are transparent about limitations, adding to the credibility of the work.

**Summary Of The Paper:**

This paper presents MRC (Medical Recursive Confidence), a novel framework for recursive uncertainty modeling in medical diagnosis. The approach combines hierarchical confident learning with uncertainty-aware GANs to propagate diagnostic confidence through layered decision processes. The goal is to manage uncertainty more systematically in clinical decision-making pipelines.

**Weaknesses:**

1. It's unclear whether the framework can handle multi-modal fusion, or if it is limited to single-modality pipelines.

---

### Official Review · Reviewer_1RNE · 2025-07-18
**Review of   Recursive Confidence Propagation in Medical Diagnosis: A Hierarchical Uncertainty Framework Using Confident Learning and GANs**

**Confidence:** 3
**Clarity Of Writing:** fair
**Clinical Significance:** good
**Methodological Novelty:** good
**Overall Rating:** 6

**Experiments And Results:**

good

**Questions For The Authors:**

The introduction claims that medical diagnosis involves “cascading layers of uncertainty,” and a “unique form of diagnostic uncertainty recursion” is introduced as a novel insight, but is unaccompanied by references to either cognitive models of clinical reasoning or prior research in hierarchical medical inference.

How sensitive is MRC to errors at the lowest level (e.g., measurement noise)?

**Strengths:**

The paper introduces a structured approach that mirrors real clinical workflows. The recursive modeling of diagnostic confidence is well-formulated.

The paper presents a comprehensive evaluation:
- Evaluated on three diverse datasets.
- Uses multiple metrics (accuracy, ECE, recursive consistency, AUCAC).
- Ablation studies and physician evaluations add credibility to the empirical claims.

Many real diagnoses are made over time. Recursive uncertainty should also extend to time-sequential reasoning (e.g., evolving confidence over a hospital stay), which is acknowledged as future direction.

**Summary Of The Paper:**

This paper introduces the Medical Recursive Confidence (MRC) framework, which models recursive uncertainty propagation in hierarchical medical diagnostic processes. The authors argue that existing uncertainty quantification methods treat confidence as a flat (single-layer) attribute, while real-world medical reasoning involves cascading uncertainties from raw measurements to final diagnoses. MRC integrates confident learning (to handle noisy labels) and uncertainty-aware GANs (to model high-dimensional confidence distributions) to capture this multi-level dependency.

**Weaknesses:**

The Introduction of the paper is conceptually promising but methodologically weak. It lacks the necessary academic rigor in the form of citations, supporting evidence, and literature engagement to justify its claims. The Introduction must (a) situate the work in the context of existing literature, (b) substantiate claims of novelty and necessity, and (c) clearly define the limitations of prior work that this paper seeks to overcome.

The citation numbering in the paper is out of sync with the reference list, undermining its scholarly integrity. For instance, [6] in the related works refers to Gal and Ghahramani's Bayesian Deep Learning work, but in the reference list, [6] is A. Kendall and Y. Gal, not Gal and Ghahramani. Later references appear to drift or be misnumbered, making it difficult to follow or verify claims. Additionally, there is no citation for Heckerman et al.

While the GAN component improves calibration, it's unclear why a GAN is necessary versus a simpler probabilistic decoder or variational model.